# Dropped Head Syndrome: The Importance of Neurophysiology in Distinguishing Myasthenia Gravis from Parkinson’s Disease

**DOI:** 10.3390/biomedicines12081833

**Published:** 2024-08-12

**Authors:** Marilena Mangiardi, Alessandro Magliozzi, Carlo Colosimo, Luca Marsili

**Affiliations:** 1San Camillo-Forlanini Hospital, 00152 Rome, Italy; marilenamangiardi@gmail.com; 2Campus Biomedico University Hospital, 00128 Rome, Italy; a.magliozzi@unicampus.it; 3Department of Neurology, Santa Maria University Hospital, 05100 Terni, Italy; 4Gardner Family Center for Parkinson’s Disease and Movement Disorders, Department of Neurology, University of Cincinnati, Cincinnati, OH 45219, USA

**Keywords:** dropped head syndrome, antecollis, myasthenia gravis, Parkinson’s disease, neurophysiology

## Abstract

Dropped head syndrome (DHS) is characterized by severe forward flexion of the cervical spine due to an imbalance in neck muscle tone. This condition can be linked to various neuromuscular diseases, including myasthenia gravis (MG). On the other hand, Parkinson’s disease (PD) patients may show a clinically indistinguishable picture named antecollis, which is caused by increased axial tone, but without muscle weakness. Differentiating between DHS and antecollis is crucial due to their distinct treatment requirements. We present the case of a 71-year-old White male with a one-month history of severe neck flexion, mild dysphagia, and dysphonia. His medical history included diabetes mellitus, coronary artery disease, arterial hypertension, and mild cervical spondylosis. Neurological examination revealed features of Parkinsonism, including hypomimia, asymmetric rigidity, and reduced arm swing. There was significant weakness in his neck extensor muscles, with no signs of ptosis or diplopia. Brain/spine MRI scans were unremarkable, but electromyography showed a reduced compound muscle action potentials amplitude in repetitive nerve stimulation, consistent with MG. High-titer acetylcholine receptor antibodies confirmed the diagnosis. Treatment with pyridostigmine (60 to 120 mg/day) and plasma exchange (daily, for five consecutive days) improved the patient’s general condition and neck posture. Concurrently, the patient was diagnosed with PD based on established clinical criteria and improved with carbidopa/levodopa therapy (up to 150/600 mg/daily). This case highlights the rare co-occurrence of MG and PD, emphasizing the need for thorough clinical, neurophysiological, and laboratory evaluations in complex DHS presentations. Managing MG’s life-threatening aspects and addressing PD symptoms requires a tailored approach, showcasing the critical role of neurophysiology in accurate diagnosis and effective treatment.

## 1. Introduction

Dropped head syndrome (DHS) [1] is an abnormal forward flexion of the cervical spine resulting in a distinctive, and often debilitating, neck posture [2]. This condition primarily arises due to an imbalance in the tone of the neck muscles, where the flexors overpower the extensor muscles [1]. DHS can be caused by a variety of neuromuscular diseases causing neck muscle weakness, such as myasthenia gravis (MG), motor neuron disease, and myopathies [3,4]. In contrast, Parkinson’s disease (PD) can present with a clinically identical neck posture, named antecollis, which is caused by an increased axial tone of the anterior cervical muscles but without muscle weakness [5,6,7]. Hence, several differentials should be evaluated when facing DHS and/or antecollis [3].

To complicate matters, the co-occurrence of MG and PD in the same patient, even if rare, has been reported previously [3]. Both conditions, in fact, can contribute to abnormal neck posturing, but DHS and antecollis require distinct diagnostic approaches and treatment regimens. With the present report, we describe a rare case of a patient presenting with DHS caused by new-onset MG, who was diagnosed with PD at the same time. This dual diagnosis, particularly in the absence of previous neurological complaints, highlights the importance of thorough clinical, neurophysiological, and laboratory evaluations in patients with DHS. The concurrent presence of MG and PD, while rare, necessitates a careful and prioritized approach to treatment, emphasizing the need for the immediate management of life-threatening conditions associated with MG but at the same time without neglecting PD. By documenting this case, we aim to highlight the intricate relationship between MG and PD in the manifestation of DHS and the essential role of neurophysiology in distinguishing between these conditions to guide appropriate therapeutic strategies.

## 2. Detailed Case Description

A 71-year-old White male presented to our clinic with a primary complaint of a subtle onset of severe neck flexion, which had started several months prior and progressively worsened (Appendix A Segment 1 shows DHS, with the patient struggling to keep his head elevated and straight). This condition was accompanied by mild dysphagia and dysphonia. His past medical history was relevant for type II diabetes mellitus, coronary artery disease, arterial hypertension, and mild cervical spondylosis. Notably, his family history revealed that his father had PD (onset in his late 60 s). He did not drink coffee or alcoholic beverages, and he did not smoke. Overall, he was in good health and did not take medications nor have any relevant sleep disorders.

The neurological examination showed Parkinsonian features such as hypomimia, mild bradykinesia, and asymmetric rigidity, associated with a shuffling gait. Importantly, there were no signs of ptosis or diplopia. The detailed examination of the neck muscles revealed significant weakness in the neck extensors, with a muscle strength rating of 2/5 according to the Medical Research Council’s scale. All other muscle groups were rated 5/5, suggesting normal strength. Cognition was intact.

Brain and cervical spine MRI scans were unremarkable, ruling out possible structural abnormalities. Electromyography (EMG) showed a reduction in the amplitude and area of compound muscle action potentials (CMAP) [8] between the first and fourth stimuli during repetitive nerve stimulation of the neck extensor muscles. This pattern was consistent with abnormal neuromuscular transmission. The needle EMG exam was not indicative of myopathic or neuropathic features. Subsequent diagnostic tests revealed high-titer (8 nmol/L) positivity for acetylcholine receptor-binding antibodies (normal values < 0.40 nmol/L) [9], a hallmark of MG. Other laboratory tests, including levels of creatine kinase, aldolase, thyroid-stimulating hormone, muscle-specific kinase antibodies, and a standard autoimmune/encephalopathic panel, were within normal limits. A contrast-enhanced chest CT scan was negative for thymoma.

Based on these findings, the DHS was attributed to MG, and treatment with pyridostigmine ranging from 60 to 120 mg/day was promptly started and followed by plasma exchange (daily, for five consecutive days), deemed necessary to improve his general condition, including his neck posture. Simultaneously, the patient’s neurological examination showed a Parkinsonian syndrome. In the absence of red flags for atypical Parkinsonism, the most plausible diagnosis was of PD, according to the current clinical diagnostic criteria [6]. Hence, the patient was started on carbidopa/levodopa, titrated up to 25/100 mg two tablets three times daily. This regimen resulted in an improvement of his mild Parkinsonian symptoms, including reduced stiffness and a faster gait, now visible at the 12-month follow-up (Appendix A Segment 2 shows the patient walking at a fast pace, able to turn around, but with some residual Parkinsonian features, such as reduced arm swing and shuffling gait, despite treatment).

## 3. Discussion

MG is an autoimmune disorder affecting neuromuscular transmission, leading to muscle weakness and fatigue [9]. The primary cause of altered neurotransmission in MG is the presence of autoantibodies targeting components of the neuromuscular junction (NMJ), particularly the AChRs on the postsynaptic muscle membrane [9]. Autoantibodies bind to AChRs, leading to their cross-linking and internalization [10]. This reduces the number of functional AChRs available on the muscle cell surface, diminishing the muscle’s ability to respond to the acetylcholine [11] released from the nerve terminal [12]. The binding of antibodies to AChRs activates the complement system, which leads to inflammation and damage to the postsynaptic membrane [13]. Inflammation, in turn, further decreases the number of functional AChRs and disrupts the architecture of the NMJ [14]. With fewer AChRs, the binding of ACh to the receptors is insufficient to generate a strong end-plate potential (EPP) [11]. A weak EPP fails to consistently reach the threshold required to trigger muscle action potentials, resulting in reduced muscle contraction strength [15]. The U-shaped pattern observed in the CMAPs during the repetitive nerve stimulation of MG patients reflects the initial depletion of available AChRs followed by a partial recovery due to the increased mobilization of ACh vesicles and a transient improvement in neuromuscular transmission [8]. Figure 1 represents a detailed graphical explanation of the phenomenon in healthy subjects (top) and in MG patients (bottom) for comparison, while Figure 2 represents the results of the repetitive nerve stimulation [8] performed on our patient.

DHS [1] and antecollis can both be caused by a variety of conditions, as detailed in Table 1. Our diagnostic algorithm, based on careful clinical examination, followed by a broad autoimmune/encephalopathy panel, EMG, and neuroimaging helped in achieving the correct diagnosis. It is also important to differentiate DHS and antecollis from kyphosis and other spinal abnormalities, which represent fixed deformities [3]. Differentiating between DHS due to MG [9], and antecollis due to Parkinsonism (with increased axial tone and passive extension to a normal position), is crucial [3,5]. This distinction is important because the treatments of MG and PD significantly differ (Table 2). In more detail, the new onset of DHS due to MG has been described in patients with a previous diagnosis of PD, thus suggesting that MG be considered in PD patients with an acute/subacute onset of DHS [5,16]. However, the simultaneous diagnosis of MG and PD in a patient without any known pre-existing neurological conditions is unprecedented and presents a unique clinical challenge.

In managing such cases, clinicians must carefully evaluate both conditions, prioritizing the life-threatening aspects of MG that require immediate intervention with appropriate therapies such as pyridostigmine, steroids, IVIG/plasmapheresis, and/or other immunomodulatory drugs, as well as DHS (Table 3). Distinguishing between DHS due to MG and antecollis due to Parkinsonism is essential for effective management, as their therapeutic approaches are different. In challenging diagnostic cases, the expertise of a neurophysiologist and the application of advanced neurophysiological techniques are crucial. Repetitive nerve stimulation (RNS) [8] is a valuable diagnostic tool for MG, known for its high specificity, although it has limited sensitivity. For more complex or inconclusive cases where RNS does not yield clear results, single-fiber EMG serves as an essential second-level diagnostic test (Table 4).

An interesting aspect is the possible common pathogenesis of some biological subtypes of PD and MG and/or other immune-mediated disorders [21]. Cases of PD/MG comorbidities have been reported [22,23,24,25]. Studies have suggested that in some PD subtypes, pathogenesis can be associated with neuroinflammation and/or immune-mediated mechanisms [26]. In fact, antibodies isolated from PD patients can tackle dopaminergic cells or antigens such as melanin, α-synuclein, and GM1 ganglioside, all linked to PD pathogenesis [23]. In addition, neuromelanin can activate dendritic cells and microglia, which, in turn, may favor proliferative T- and B-cell immune responses [27].

A final comment entails the management of MG and PD. Even if not related to the present case, we wanted to highlight that there is no contraindication in the concurrent use of medications to treat both conditions [28]. In more detail, several warnings are reported when using pyridostigmine and rivastigmine (frequently used in PD for cognitive deficits and also to treat orthostatic hypotension [29]) at the same time. However, based on our clinical experience, they can be used together and may have an additive effect.

The present case report has some limitations such as the short-term follow-up investigation and the lack of ioflupane 123I-FP-CIT dopamine transporter (DaT) SPECT analysis (the patient refused to undergo this exam) to further confirm the diagnosis of PD. However, we are confident in the diagnosis, based on the diagnostic criteria and the response to dopaminergic therapy. We cannot fully exclude that the patient had subtle prodromal or pre-morbid signs of PD before the onset of MG; however, he was not complaining of any other symptoms suggestive of PD before his admission to our center.

## 4. Conclusions

In conclusion, this case highlights how a patient with a subacute-onset and progressive course of DHS was ultimately diagnosed with MG based on clinical, neurophysiological, and laboratory findings. Prompt treatment initiation with pyridostigmine and plasmapheresis was crucial for managing the MG symptoms. Concurrently, a diagnosis of PD was established, further supported by the patient’s positive response to levodopa treatment. This case underscores the importance of comprehensive evaluation and tailored treatment strategies in patients presenting with complex neuromuscular and movement disorder symptoms. It also highlights the necessity for clinicians to maintain a high index of suspicion for concurrent neurological conditions, even in the absence of prior neurological complaints, to ensure timely and appropriate management. Finally, PD and MD may share, at least in part, common pathogenetic mechanisms, and this aspect should be further investigated in prospective and epidemiological studies.

## Figures and Tables

**Figure 1 biomedicines-12-01833-f001:**
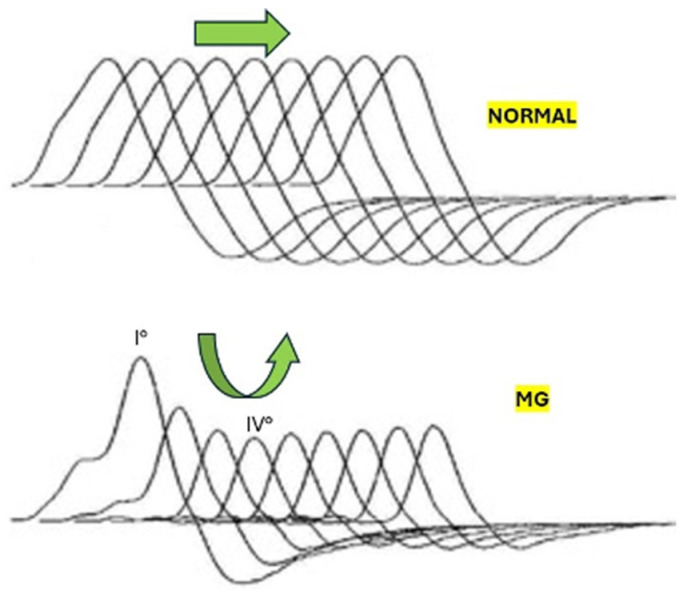
Repetitive nerve stimulation traces. The top set represents a normal condition, while the bottom set depicts a patient with myasthenia gravis. The normal repetitive nerve stimulation traces show consistent, overlapping waveforms with each subsequent stimulus. The compound muscle action potential amplitudes remain stable across all stimuli, indicating efficient neuromuscular transmission. The myasthenia gravis (MG) repetitive nerve stimulation traces (bottom set) show a decremental response, with a noticeable reduction in the amplitude of successive compound muscle action potential. There is a progressive decline in compound muscle action potential amplitude with each subsequent stimulus, characteristic of impaired neuromuscular transmission due to the reduced number of functional acetylcholine receptors at the neuromuscular junction (see progressively reduced amplitude from I° to IV°).

**Figure 2 biomedicines-12-01833-f002:**
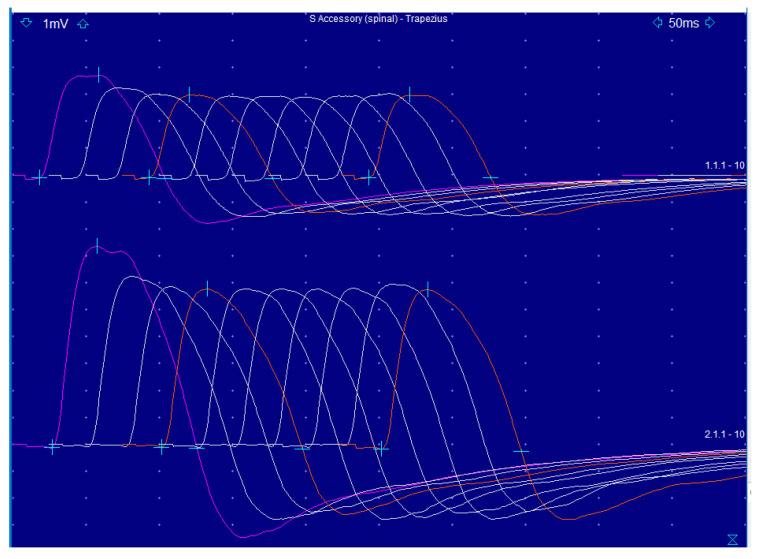
Altered repetitive nerve stimulation results in our patient. The graph shows the results of the repetitive nerve stimulation of the patient’s right spinal accessory nerve recorded from the trapezius muscle. A valuable decrease in the amplitude and area of the compound muscle action potential between the first and fourth stimuli (19% decrease in amplitude; 22% decrease in area) was observed both at rest and after 2 min of muscle activation.

**Table 1 biomedicines-12-01833-t001:** Differential diagnoses for dropped head syndrome and antecollis.

Condition	Diagnosis	Clinical Signs	Notes	References
**Dropped head syndrome**				
*Myasthenia Gravis*	EMG, Ach/mUSK antibodies	Dysphagia, dysphonia, weakness, fluctuating fatigue	Responds to pyridostigmine and plasmapheresis	[5,16]
*Motor Neuron Disease* *(ALS, PPS)* *	EMG	Hyperreflexia, Babinski, weakness, fasciculations	Can have dysphonia and dysphagia	[3]
*Myopathy* *(PM, IBM, NM, MM, CD, FSHD, Radiation-induced, AED-induced carnitine deficiency)*	Muscle biopsy	Weakness	LGMD, FSHD, IBM, other myositis, amyloid, thyroid myopathy, pay attention to epileptic patients taking valproic acid	[3,17]
*IGLON-5 Disease*	IGLON-5 antibodies	Weakness, bulbar symptoms, sleep disorders	Tauopathy, rarely responds to immune therapy, severe prognosis	[18]
**Antecollis**				
*Parkinson’s Disease*	Clinical	Bradykinesia plus at least one between tremor, rigidity, postural instability	Responds to levodopa/carbidopa	[6]
*Multiple System Atrophy*	Clinical	Dysautonomia, atypical Parkinsonism, cerebellar symptoms	More frequently displays antecollis than idiopathic PD	[7]
*Idiopathic Cervical Dystonia* *	Clinical	Sensory trick	Responds to botulinum toxin injections in anterior neck muscle groups and anticholinergic drugs	[19]
*DPPX-Encephalopathy*	DPPX antibodies	Weight loss, dysautonomia, GI symptoms	Responds to immune therapies	[20]

ALS: Amyotrophic Lateral Sclerosis; PPS, Postpolio Syndrome; PM: Polymyositis; IBM: Inclusion Body Myositis; NM: Nemaline Myopathy; MM: Mitochondrial Myopathy; CD: Carnithine Deficiency; FSHD: Facioscapulohumeral Dystrophy; LGMD: Limb–Girdle Muscular Dystrophies; AEDs: Anti-epileptic Drugs; DPPX: Anti-dipeptidyl-peptidase-like Protein; EMG, electromyography; Ach, acetylcholine receptor-binding antibodies; mUSK, muscle-specific kinase antibodies. * Genetic variants (with strong family history) of axial dystonia associated with *ANO3* gene have been reported.

**Table 2 biomedicines-12-01833-t002:** Proposed treatments for dropped head syndrome and antecollis.

Condition	Treatment	Notes
**Dropped head syndrome**		
*Myasthenia Gravis*	IVIG, PLEX, Steroids	May improve/resolve
*Motor Neuron Disease* *(ALS, PPS)*	-	Consider neck brace, treat dysphagia if associated
*Myopathy* *(PM, IBM, NM, MM, CD, FSHD, Radiation-induced, and AED-induced carnitine deficiency)*	Consider steroids in responsive forms; discontinue valproic acid in epileptic patients	Consider neck brace, physical therapy
*IGLON-5 Disease*	IVIG, PLEX, Steroids	Weak response
**Antecollis**		
*Parkinson’s Disease*	Discontinue dopamine-agonists or other recently added treatments; consider levodopa increase and muscle relaxants	Physical therapy, consider BoNT injections in anterior neck muscles
*Multiple System Atrophy*	Same as Parkinson’s disease	-
*Idiopathic Cervical Dystonia*	Consider BoNT injections in anterior neck muscles and anticholinergics	Effective, but increased risk of dysphagia
*DPPX-Encephalopathy*	IVIG, PLEX, Steroids	May improve

ALS: Amyotrophic Lateral Sclerosis; PPS, Postpolio Syndrome; PM: Polymyositis; IBM: Inclusion Body Myositis; NM: Nemaline Myopathy; MM: Mitochondrial Myopathy; CD: Carnithine Deficiency; FSHD: Facioscapulohumeral Dystrophy; LGMD: Limb–Girdle Muscular Dystrophies; AEDs: Anti-epileptic Drugs; DPPX: Anti-dipeptidyl-peptidase-like Protein; IVIG: Intravenous Immunoglobulins; PLEX: Plasma Exchange; BoNT: Botulinum Neurotoxin.

**Table 3 biomedicines-12-01833-t003:** Available treatments for myasthenia gravis.

Treatment	Description	Mechanism of Action	Notes
*Acetylcholinesterase inhibitors*	Medications that improve communication between nerves and muscles	Inhibit breakdown of acetylcholine, increasing its availability at neuromuscular junctions	First-line treatment; includes pyridostigmine. May cause gastrointestinal side effects
*Corticosteroids*	Anti-inflammatory drugs used to suppress immune response	Reduce autoantibody production and inflammation	Effective for moderate to severe MG; long-term use can cause significant side effects like osteoporosis and diabetes
*Immunosuppressants*	Drugs that suppress the immune system to reduce autoantibody production	Target immune cells to decrease autoantibody levels	Include azathioprine, mycophenolate mofetil, and cyclosporine. Regular monitoring required due to risk of infection and other side effects
*Plasma exchange*	Procedure aiming to remove antibodies from the blood.	Direct removal of circulating autoantibodies	Used in acute exacerbations or crisis situations. Short-term relief; risk of complications like infection or blood clots
*Intravenous immunoglobulins (IVIG)*	Infusion of antibodies from donated blood to modulate the immune system	Provides normal antibodies to alter immune response and neutralize autoantibodies	Used in acute exacerbations or as maintenance therapy. High cost and risk of allergic reactions
*Thymectomy*	Surgical removal of the thymus	Potentially reduces the production of autoantibodies by removing a source of abnormal immune response	May lead to remission or reduced medication requirement; most effective in patients with thymoma or early-onset MG
*Monoclonal antibodies*	Targeted biologic therapy that specifically inhibits components of the immune system	Block specific immune pathways, such as complement activation or B-cell function	Include rituximab and eculizumab. Used in refractory cases; expensive and require careful monitoring
*Novel/experimental therapies*			
*- Efgartigimod*	Neonatal Fc receptor (FcRn) antagonist	Reduces levels of pathogenic IgG antibodies	Demonstrates efficacy in clinical trials; may cause headache, upper respiratory infections
*- Rozanolixizumab*	Humanized monoclonal antibody targeting the neonatal FcRn	Lowers circulating IgG autoantibodies by blocking FcRn	Under investigation; shown promising results in reducing symptoms. Infusion-related reactions possible

MG, myasthenia gravis; FcRn, neonatal Fc receptor antagonist.

**Table 4 biomedicines-12-01833-t004:** Distinctive repetitive nerve stimulation findings among neuromuscular disorders.

Condition	RNS Findings	Clinical Features	Diagnostic Tests	Notes
*Myasthenia Gravis (MG)*	Decremental response in CMAP amplitude, often U-shaped	Fluctuating muscle weakness, fatigability, dysphagia, dysphonia	Acetylcholine receptor antibodies, SFEMG	High specificity with RNS; SFEMG for inconclusive cases
*Lambert-Eaton Myasthenic Syndrome (LEMS)*	Incremental response in CMAP amplitude upon high-frequency stimulation	Proximal muscle weakness, autonomic dysfunction	Voltage-gated calcium channel antibodies	CMAP amplitude increases after exercise or high-frequency stimulation
*Amyotrophic Lateral Sclerosis (ALS)*	Reduced CMAP amplitude, no decremental pattern	Progressive muscle weakness, atrophy, fasciculations	EMG, clinical examination	ALS affects both upper and lower motor neurons
*Botulism*	Decremental response in CMAP amplitude, improvement with rapid RNS	Acute onset of muscle weakness, cranial nerve involvement	Toxin detection in serum, stool, or food	Requires prompt diagnosis and treatment
*Congenital Myasthenic Syndromes (CMS)*	Decremental response in CMAP amplitude	Early onset muscle weakness, often with ocular involvement	Genetic testing, SFEMG	Genetic forms of neuromuscular junction disorders

CMAP: compound muscle action potential; RNS: repetitive nerve stimulation; SFEMG: single-fiber EMG.

## Data Availability

The original contributions presented in the study are included in the article/Appendix A, further inquiries can be directed to the corresponding author/s.

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
