# Peer review of "Dropped Head Syndrome: The Importance of Neurophysiology in Distinguishing Myasthenia Gravis from Parkinson’s Disease"

_biomedicines, 2024, doi:10.3390/biomedicines12081833_

Round 1

Reviewer 1 Report

Comments and Suggestions for Authors

The authors described the case report entitled "Dropped head syndrome: The importance of neurophysiology in distinguishing myasthenia gravis from Parkinson’s disease". Parkinson disease and myasthenia gravis share many motor and nonmotor clinical features including fatigue, weakness, dysarthria, dysphagia, and diplopia, which may mask or delay the diagnosis of MG. Hence, diagnosis of PD and MG is of great significance. In the present study the authors described the case of co-occurrence of PD and MG. I have some suggestions and clarifications regarding this manuscript. Please find the comments below;

1. Authors may first define "Dropped head syndrome" in the abstract itself.

2. In the abstract, authors may include the dose of treatment (for every drug)

3. "The primary cause of altered neurotransmission in MG is the presence of autoantibodies targeting components of the neuromuscular junction, particularly the AChRs on the postsynaptic muscle membrane. Autoantibodies bind to AChRs, leading to their cross-linking and internalization [9]. This reduces the number of functional AChRs available on the muscle cell surface, diminishing the muscle’s ability to respond to acetylcholine (ACh) released from the nerve terminal. Binding of antibodies to AChRs activates the complement system, which leads to inflammation and damage to the postsynaptic membrane. Inflammation, in turn, further decreases the number of functional AChRs and disrupts the architecture of the NMJ. With fewer AChRs, the binding of ACh to the receptors is insufficient to generate a strong end-plate potential (EPP). A weak EPP fails to consistently reach the threshold required to trigger muscle action potentials, resulting in reduced muscle contraction strength. The U-shaped pattern observed in the CMAPs during repetitive nerve stimulation of MG patients, reflects the initial depletion of available AChRs followed by a partial recovery due to increased mobilization of ACh vesicles and a transient improvement in neuromuscular transmission". The authors should incoporate appropriate citations in these text.

4. Figure can be renamed as Figure 1

5. Is there any proper literature to correlate the DHS and Antecollis with any of the specific neuromuscular disease.

Author Response

General comment: The authors described the case report entitled "Dropped head syndrome: The importance of neurophysiology in distinguishing myasthenia gravis from Parkinson’s disease". Parkinson disease and myasthenia gravis share many motor and nonmotor clinical features including fatigue, weakness, dysarthria, dysphagia, and diplopia, which may mask or delay the diagnosis of MG. Hence, diagnosis of PD and MG is of great significance. In the present study the authors described the case of co-occurrence of PD and MG. I have some suggestions and clarifications regarding this manuscript. Please find the comments below;

Response: We thank the Reviewer for the general positive overview of our case report. 

Specific comment 1. Authors may first define "Dropped head syndrome" in the abstract itself.

            Response: We thank the Reviewer for pointing out this important aspect. We have now defined the dropped head syndrome (DHS) in the abstract and in the introduction of the manuscript (Page: 1; Lines: 17, 39).

Specific comment 2. In the abstract, authors may include the dose of treatment (for every drug).

            Response: We have now added the details of treatment as required by the Reviewer (Page: 1; Lines 26-29).

Specific comment 3. "The primary cause of altered neurotransmission in MG is the presence of autoantibodies targeting components of the neuromuscular junction, particularly the AChRs on the postsynaptic muscle membrane. Autoantibodies bind to AChRs, leading to their cross-linking and internalization [9]. This reduces the number of functional AChRs available on the muscle cell surface, diminishing the muscle’s ability to respond to acetylcholine (ACh) released from the nerve terminal. Binding of antibodies to AChRs activates the complement system, which leads to inflammation and damage to the postsynaptic membrane. Inflammation, in turn, further decreases the number of functional AChRs and disrupts the architecture of the NMJ. With fewer AChRs, the binding of ACh to the receptors is insufficient to generate a strong end-plate potential (EPP). A weak EPP fails to consistently reach the threshold required to trigger muscle action potentials, resulting in reduced muscle contraction strength. The U-shaped pattern observed in the CMAPs during repetitive nerve stimulation of MG patients, reflects the initial depletion of available AChRs followed by a partial recovery due to increased mobilization of ACh vesicles and a transient improvement in neuromuscular transmission". The authors should incoporate appropriate citations in these text.

            Response 3: Following the Reviewer’s suggestions, we have now added the required references where missing (Page: 3; Lines: 100-119).

Specific comment 4. Figure can be renamed as Figure 1

            Response 4: The figure has been renamed as Figure 1.

Specific comment 5. Is there any proper literature to correlate the DHS and Antecollis with any of the specific neuromuscular disease.

            Response 5: We thank the Reviewer for highlighting this interesting aspect. Both antecollis and DHS are clinical pictures associated with different patterns of head and neck muscular involvement, but to the best of our knowledge, are not pathognomonic of any specific neuromuscular (or other) diseases. We agree that a more detailed study on this topic would be fascinating, however, this is out of the topic of our case report. This is definitely something we are interested in further studying in the future, as it belongs to our area of scientific interest.

Reviewer 2 Report

Comments and Suggestions for Authors

Thank you for inviting me to review this Case Report. This study was mainly focused on dropped head syndrome. The topic was of a certain significance nowadays. The submission fell within the scope of Biomedicines. The reviewer suggested a Minor Revision for this submission. Detailed comments:

1.       Demographic information of the patient should be provided.

2.       The reviewer appreciated the recording of the video. To better take advantages of the video, it was recommended to provide more literal description of it in the text.

3.       More information should be provided regarding the Figure. What did the curves mean? Were they obtained from the patient? Relevant methodologies and analyses should be strengthened.

4.       After documenting this case, did you have some suggestions for the PD guidelines? Please discuss.

Author Response

General comment: Thank you for inviting me to review this Case Report. This study was mainly focused on dropped head syndrome. The topic was of a certain significance nowadays. The submission fell within the scope of Biomedicines. The reviewer suggested a Minor Revision for this submission.

Response: We thank the Reviewer for the positive general feedback on our study.

Detailed comments:

Comment 1.       Demographic information of the patient should be provided.

      Response: According to the Reviewer’s comment, we have now added the demographic/clinical information of the patient (Page: 2; Lines: 62-70).

Comment 2.       The reviewer appreciated the recording of the video. To better take advantages of the video, it was recommended to provide more literal description of it in the text.

      Response: We thank the Reviewer for the positive feedback on the video recording and, accordingly, we have now added a detailed description of the video within the main text (Page: 2; Lines 64-65, and Pages:2-3; Lines:97-99).

Comment 3.       More information should be provided regarding the Figure. What did the curves mean? Were they obtained from the patient? Relevant methodologies and analyses should be strengthened.

      Response: We thank the Reviewer for this insightful comment. We have now improved the description of the Figure 1 in the text (Page: 3; Lines: 117-119) as well as in the figure legend (Page: 3; Line 126). Also, as required by the Editor and to improve clarity, we have now added a second Figure (Figure 2), with the detailed description of the results of the repetitive stimulation’s result in our patient.

Comment 4.       After documenting this case, did you have some suggestions for the PD guidelines? Please discuss.

      Response: We agree with the Reviewer that a better knowledge of postural abnormalities in PD would be crucial. As a matter of fact, we have dedicated several scientific efforts in the field of postural abnormalities in parkinsonian syndromes, and this is documented by the track record of the senior authors of the present manuscript, thus supporting our interest in the field. The current PD diagnostic criteria state that disproportionate antecollis is a “red flag” for atypical parkinsonism (Postuma et al. MDS clinical diagnostic criteria for Parkinson's disease. Mov Disord. 2015 Oct;30(12):1591-601), and this is an important aspect to outline. However, in view of the renewed interest towards a biological definition of PD (See also Simuni et al. A biological definition of neuronal α-synuclein disease: towards an integrated staging system for research. Lancet Neurol. 2024 Feb;23(2):178-190), we believe that a deeper knowledge of the biological common pathways underlying diseases like PD and MG would be crucial in better understanding disease progression and, possibly, also to identify new disease-modifying treatment strategies. Hence, we have added a concluding remark to the conclusion section, to highlight this important aspect (Page: 8; Lines: 215-217).

Reviewer 3 Report

Comments and Suggestions for Authors

Dear Authors,

The manuscript consists of a well-documented description of an interesting clinical case. 

The manuscript is accompanied by a discussion of the differential diagnoses for dropped head and antecollis (Table 1). I would like to point out that a further condition of dropped head syndrome, observed with a fair amount of frequency by neurologists who – like myself – deal with clinical epilepsy, is represented by patients with chronic epilepsy who receive a robust polytherapy (Brázdil, M., Fojtíková, D., Košt’álová, E., Bareš, M., Kuba, R., Pažourková, M., & Rektor, I. Dropped head syndrome in severe intractable epilepsies with mental retardation. Seizure, 2005, 14(4), 282-287), a condition that can be assumed to be included among the myopathies caused by secondary carnitine deficiency.

Best regards. 

Author Response

Comment: Dear Authors,

The manuscript consists of a well-documented description of an interesting clinical case. 

The manuscript is accompanied by a discussion of the differential diagnoses for dropped head and antecollis (Table 1). I would like to point out that a further condition of dropped head syndrome, observed with a fair amount of frequency by neurologists who – like myself – deal with clinical epilepsy, is represented by patients with chronic epilepsy who receive a robust polytherapy (Brázdil, M., Fojtíková, D., Košt’álová, E., Bareš, M., Kuba, R., Pažourková, M., & Rektor, I. Dropped head syndrome in severe intractable epilepsies with mental retardation. Seizure, 2005, 14(4), 282-287), a condition that can be assumed to be included among the myopathies caused by secondary carnitine deficiency. 

Best regards.

            Response:  We thank the Reviewer for the positive feedback provided on our manuscript. We have now added the “AED-induced carnitine deficiency” among the myopathy category in Table 1, and in the new Table 2 and we added the interesting supporting reference as well (new Ref# 17).